# FLOWING WITH PRECISION: RECTIFIED FLOW IMAGE EDITING WITH TRAJECTORY AND FREQUENCY GUIDANCE

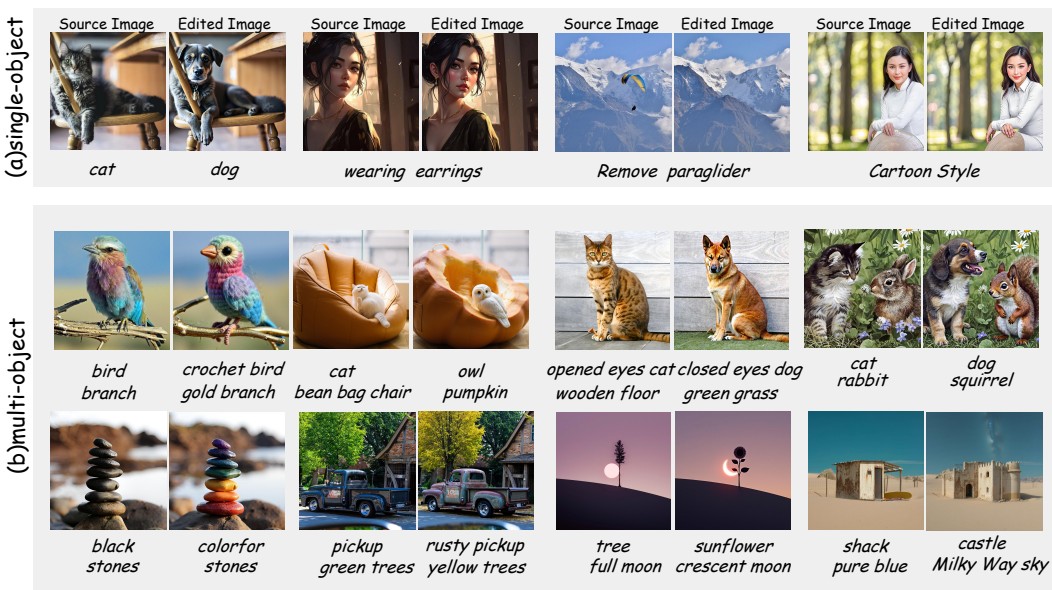

Figure 1: Our method is designed to modify high level attributes to match the prompt, all while maintaining the core structure and background of the source image.

## ABSTRACT

Rectified flow text-to-image models have shown remarkable progress. However, editing complex scenes containing multiple objects remains challenging due to semantic entanglement and structural inconsistency. To address this, we propose a dual-domain framework that jointly refines temporal editing trajectories and adapts frequency domain. First, we design a **Starting Point Optimization (SPO) strategy**, which intelligently determines the optimal editing starting point based on the structural complexity of different images. Second, we introduce a **Trajectory Optimization (TO) strategy**. In the time domain, it performs semantic-aware vector orthogonalization to suppress source bias while preserving target semantics. In the frequency domain, it adaptively re-weights high and low frequency residuals according to stage-specific spectral characteristics. Furthermore, we leverage the frequency-aware capabilities of MM-DiT to dynamically inject structural priors from the source image at different denoising steps. Our method allows users to add, replace, or modify multiple objects, making it highly efficient for editing complex scenes. Experiments show that our method significantly outperforms existing methods for image editing and achieving higher user preference in human evaluations.

## 1 INTRODUCTION

The goal of image editing is to align a region of interest with text prompt while preserving the non-edited areas. Recently, Rectified Flow (RF) models (Labs, 2024) have demonstrated superior performance over diffusion models (DMs) (Rombach et al., 2022; Tang et al., 2023; Wang et al., 2024; Song et al., 2022) in both image quality and text alignment, by leveraging flow matching (Lipman et al., 2023; Liu et al., 2022) and a multi-modal diffusion transformer (MM-DiT) (Esser et al., 2024; Peebles and Xie, 2023; Huang et al., 2024). However, their effectiveness is limited when it comes to fine-grained editing of specific, detailed regions within images containing multiple objects or complex scenes. In practice terms, we aim to design a powerful and effective framework that excels at image editing across a diverse range of image types.

The challenge in image editing is to precisely edit multi-object at specific locations within an image. Existing text-guided image editing methods that leverage flow models (Wang et al., 2025; Deng et al., 2024; Avrahami et al., 2025) operate by inverting the source image into a latent space and then performing conditional denoising. However, such methods often lead to significant deviations between the edited and original images (Xie et al., 2025; Kulikov et al., 2025). While some approaches utilize attention modification to enhance control (Xu et al., 2024; Tewel et al., 2024; Lv et al., 2025), they often face challenges. While attention-based methods are good at preserving the original image structure, this strong control can also restrain the editing strength. When multiple objects are present, these approaches often suffer from appearance leakage (Zhang et al., 2025; Sun et al., 2025). Current multi-object editing methods (Zhu et al., 2025a; Yang et al., 2024; Huang et al., 2025) rely on masks, such methods often struggles to precisely bind specific attributes to their geometrically defined regions, which can lead to inaccurate edits or content leakage.

To overcome the above challenges, we revisit the editing process. Unlike DM, RF allows the latent noise to be inferred at each time step through linear interpolation. FlowEdit builds an ODE between the source and target images without inversion. Inspired by this, we propose an inversion-free method that modifies specific regions directly on the source image at each step. We clearly divide the editing process in the noise space into three distinct stages: the **Chaos Phase**, the **Layout Phase**, and the **Refinement Phase**. Our key finding is that beginning the edit too early can compromise the source image's structural integrity, whereas starting too late may result in an ineffective edit. This is further complicated by the fact that the structural complexity of different images, leading to a non-uniform end point for the Chaos phase. To solve this issue, we introduce **Starting Point Optimization (SPO)**, a method that adaptively determines the optimal editing start point by calculating the low frequency Mean Squared Error (MSE) between the source and target images.

Although the SPO strategy enhance structural fidelity, it still suffers from obvious visual artifacts and insufficient editing strength. To this end, we propose a trajectory optimization strategy. This strategy decomposes the editing direction into two orthogonal components, cross-cue and cross-track, in the time domain, and eliminates its projection in the cross-cue direction by orthogonalizing the cross-track term. In the frequency domain, we employ dynamic frequency weighting to adaptively adjust the editing strength based on the frequency characteristics of each denoising stage. Furthermore, we leverage the frequency-aware properties of MM-DiT. Instead of applying the same strategy to all attention layers (Feng et al., 2025), we select and utilize specific attention layers at different diffusion stages to inject corresponding source image features.

In summary, our main contributions are as follows:

(a) We propose a novel **SPO** strategy that adaptively selects the optimal editing onset for different images based on their structural complexity.

(b) We introduce a **Trajectory Optimization** method. This method performs optimization in both time and frequency domains. Additionally, it selectively injects structural features from the source image into appropriate attention layers of the MM-DiT, based on the current denoising step.

(c) We evaluate our method across various editing tasks,as shown in Figure 1. Extensive experiments demonstrate that our method significantly outperforms state-of-the-art baselines in single-object editing tasks and achieves competitive results in multi-object editing scenarios.

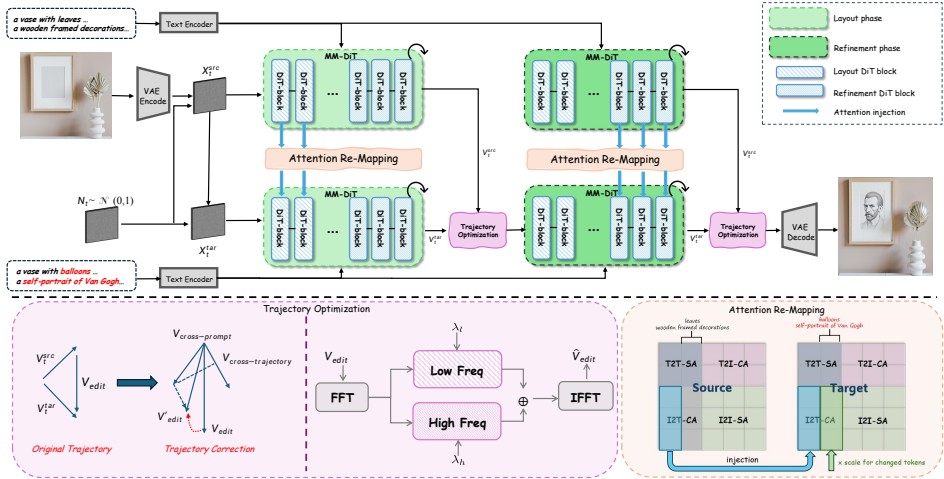

Figure 2: **Pipeline overview of our method**. (a) **Trajectory Optimization**. $\lambda_{\text{type}}(t_i) = 1 + \alpha\left(1 - \frac{\sum_{(k_x,k_y)\in\mathcal{R}_{\text{type}}}|U_{t_i}(k_x,k_y)|}{\sum_{(k_x,k_y)}|U_{t_i}(k_x,k_y)|}\right)$, where type $\in \{\text{low}, \text{high}\}$. (b) **Token mapping** based cross-attention injection,scale takes different values at different phases.

## 2 RELATED WORK

### 2.1 INVERSION FOR IMAGE EDITING

Image editing methods fall into two main categories: inversion-based and inversion-free. Inversion methods obtain the initial noise by iteratively adding predicted noise (Brack et al., 2024; Deutch et al., 2024). Null-text Inversion (Mokady et al., 2022) achieves a more precise inverse recovery by training null-text embeddings. Negative Prompt Inversion (Miyake et al., 2024) replaces the negative prompt with the source prompt. RF models still reverse the ODE to gradually add noise. RF-Inversion (Rout et al., 2024) constructs a controlled ODE through source image and noise interpolation. RF-solver (Wang et al., 2025) and FireFlow (Deng et al., 2024) introduce second-order Taylor expansions to reduce reconstruction errors. Early inversion-free methods such as SDEdit (Meng et al., 2022) strike a balance between realism and fidelity by adding moderate noise, Delta Denoising Score (Hertz et al., 2023) refines text-guided edits by subtracting noise gradients using a reference-guided image-text pair. Infedit (Xu et al., 2023) uses a special variance schedule such that the denoising step takes the same form as multi-step consistency sampling. Recently, FlowEdit (Kulikov et al., 2025) and FlowAlign (Kim et al., 2025) circumvents inversion by constructing a direct flow drive between the source and target images.

### 2.2 MULTI-OBJECT IMAGE EDITING

Balancing structural preservation and semantic alignment in multi-object editing remains a challenging task. Early methods leveraged U-Net based DMs (Jiang et al., 2025; Simsar et al., 2024) focusing on attention modification. P2P (Wang et al., 2022) manipulated cross-attention maps for feature-prompt alignment, PnP (Tumanyan et al., 2022b) injected aligned internal controls, and MastCtrl (Cao et al., 2023) achieved edits while preserving overall texture and consistency. Recent methods (Sanjyal, 2025) introduce refined control strategies. OIR (Yang et al., 2024) separates editing pairs with masks and distinct inversion steps, LoMOE (Chakrabarty et al., 2024) restricts edits to specified mask regions through a multi-diffusion process, and Paralleledits (Huang et al., 2025) employs parallel branches for a editing strategy. While RF models primarily (Rombach et al., 2022; Labs, 2024) utilize the MM-DiT architecture, this domain remains underexplored. Existing methods (Deng et al., 2024; Zhu et al., 2025b; Avrahami et al., 2025) adapt attention injection to preserve fidelity, while others (Xu et al., 2025) use Adaln to manipulate features for control. The MM-DiT architecture is an area still requiring deeper investigation.

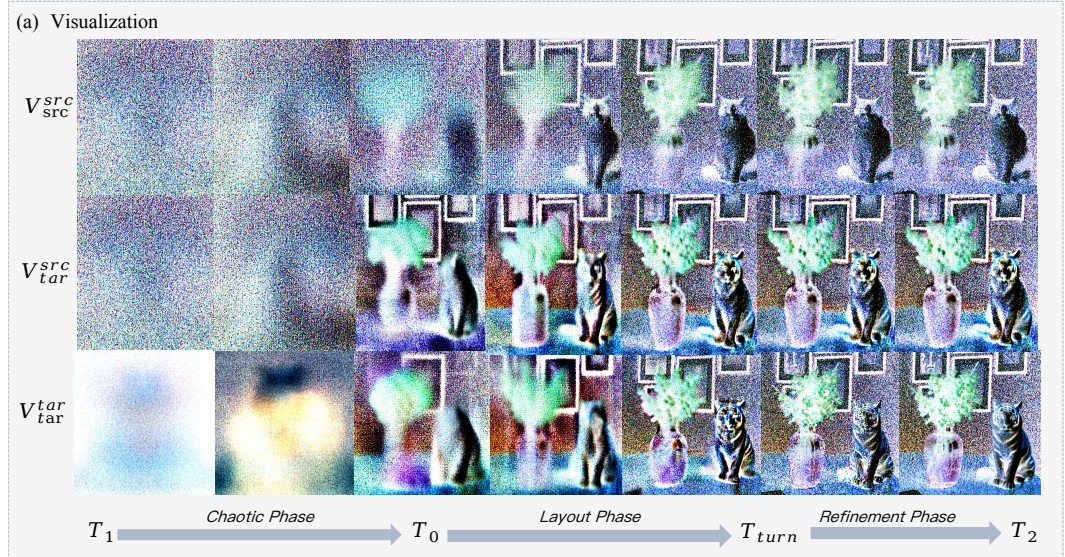

(a) Visualization

$V_{\mathrm{src}}^{src}$

$V_{tar}^{src}$

$V_{tar}^{tar}$

$T_1$ — Chaotic Phase → $T_0$ — Layout Phase → $T_{turn}$ — Refinement Phase → $T_2$

Figure 3: **Analyse on intermediate results**. In the early stages of the denoising process, predictions are mainly guided by textual cues, and $V_{tar}^{\mathrm{src}}$ and $V_{src}^{\mathrm{src}}$ have high similarity. Later, image comes into play, and $V_{tar}^{\mathrm{src}}$ and $V_{src}^{\mathrm{src}}$ have high similarity.

## 3 METHOD

Our method is able to achieve semantic alignment while effectively maintaining the consistency of image structure. Our method mainly consists of three parts: (i) based on the analysis of frequency changes during denoising, We propose a novel **Start Point Optimization (SPO)** strategy; (ii) introduce a **Frequency aware trajectory optimization** method; (iii) introduces our feature injection process along with attention adaptation methods to enhance editability. An overview of our method is presented in Fig. 2.

### 3.1 PRELIMINARY

#### 3.1.1 RECTIFIED FLOW BASED MODELS.

Rectified flow models learn the probability paths between two distributions. Specifically, they *linearly interpolate* between two observed distributions $\mathbf{x}_0 \sim p_0$ and $\mathbf{x}_1 \sim p_1$, and model such probability transport paths using the following ordinary differential equation (ODE):

$$\mathbf{x}_t = t\mathbf{x}_1 + (1 - t)\mathbf{x}_0, \quad t \in [0, 1], \tag{1}$$

$$d\mathbf{x}_t = v_\theta(\mathbf{x}_t, t)\, dt, \quad \mathbf{x}_0 \sim p_0, \quad t \in [0, 1], \tag{2}$$

where $v_\theta(\mathbf{x}_t, t)$ denotes a time-aware velocity field governing the transport dynamics. The training objective is to directly regress the velocity field using the least-squares loss:

$$\mathcal{L} = \mathbb{E}_{t \sim \mathcal{U}[0,1],\, \mathbf{x}_1 \sim p_1} \left[ \|(\mathbf{x}_1 - \mathbf{x}_0) - v_\theta(\mathbf{x}_t, t)\|^2 \right]. \tag{3}$$

#### 3.1.2 INVERSION-FREE TEXT-BASED EDITING.

In text-based image editing using flow models, we aim to translate a source image $X^{\mathrm{src}}$ to a target image $X^{\mathrm{tar}}$ based on the text description of each image or the editing instruction. In particular, such translation can be represented through a linear conditional flow between two image distributions:

$$X_t^{\mathrm{edit}} = X_t^{\mathrm{tar}} - X_t^{\mathrm{src}} + X_0^{\mathrm{src}}. \tag{4}$$

Consequently, we can simulate the ODE for image editing by

$$\frac{dX_t^{\mathrm{edit}}}{dt} = V_{\mathrm{edit}}(t) = v(X_t^{\mathrm{tar}}, t, c_{\mathrm{tar}}) - v(X_t^{\mathrm{src}}, t, c_{\mathrm{src}}). \tag{5}$$

### 3.2 OBSERVATIONS ON PHASED EDITING IN NOISE SPACE

Previous work (Yu et al., 2023; Bao et al., 2025) have shown that diffusion models naturally emphasize different frequency components at different sampling timesteps. Inspired by this, we analyze intermediate results during the editing process at different denoising timesteps, we categorize the image editing process in the noise space into three distinct stages: **Chaotic Phase**, **Layout Phase**, and **Refinement Phase**.

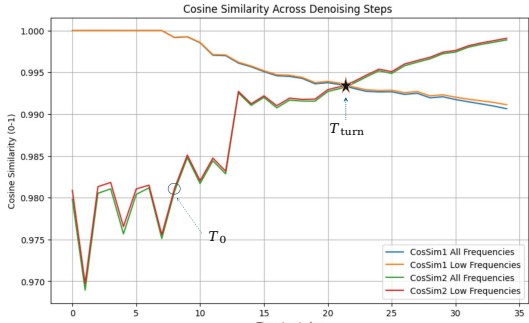

Figure 4: CosSim1 is the cosine similarity of $V_{tar}^{src}$) and $V_{tar}^{tar}$, CosSim2 is the cosine similarity of $V_{tar}^{src}$ and $V_{src}^{src}$). $T_0$ is the point where the low frequency and overall values of CosSim2 are first equal.

**Chaotic Phase:** We introduce intermediate variables $V(X_t^{tar}, t, c_{src})$. As shown in Fig. 3, during the early stage of the denoising process, the predictions are primarily guided by the text prompt, containing very little structural and semantic information about the source image. By analyzing the cosine similarity between the predictions and the target at different frequency components as shown in Fig. 4, we identify a key point $T_0$. From this point, the image information begins to guide the predictions, enabling the model to effectively reconstruct the source image's structure. Consequently, we propose a novel **SPO** strategy and define this transition point as the optimal starting timestep , which adaptively determines the starting point based on an image's characteristics.

**Layout Phase and Refinement Phase:** As illustrated in Fig. 3, after skipping the chaotic phase, $V_{src}$ is able to reconstruct the source image layout with high fidelity. The editing result then mainly depends on the accuracy of the denoising direction $V_{edit}$. We found that there is an intersection $T_{turn}$ between CosSim1 and CosSim2. Before this, it is mainly responsible for low frequency layout, and after this, it is responsible for high frequency refinement. We use this observation to delineate the **Layout Phase** and **Refinement Phase**.

### 3.3 TRAJECTORY OPTIMIZATION

#### 3.3.1 SEMANTIC AWARE VECTOR DECOUPLING

After determining the editing start point, we use Eq. (6) to build a direct ODE process between the source and target distributions. However, without explicit latent inversion, the generated $V^{edit}(t)$ vector remains heavily constrained by the source structure, which limits the overall editing strength.

$$V^{edit}(t) = \underbrace{v(X_t^{tar}, t, c_{tar}) - v(X_t^{tar}, t, c_{src})}_{\text{cross-prompt}} + \underbrace{(v(X_t^{tar}, t, c_{src}) - v(X_t^{src}, t, c_{src}))}_{\text{cross-trajectory}} \quad (6)$$

To address this, we orthogonalize the **cross-prompt** term. By retaining only the component that is independent of the **cross-trajectory** term, we avoid redundant superposition and achieve a more precise and controllable edit. Specifically, we define the cross-prompt vector as $\Delta_{ts}$ and the cross-trajectory vector as $\Delta_{ss}$. The orthogonalized cross-trajectory vector $\Delta_{ts}^{orth}$ is computed as:

$$\Delta_{ts}^{'} = \Delta_{ts} - \frac{\Delta_{ts} \cdot \Delta_{ss}}{\|\Delta_{ss}\|^2} \Delta_{ss} \quad (7)$$

where $\Delta_{ts}$ and $\Delta_{ss}$ represent the cross-prompt and cross-trajectory vectors, respectively.

$$V_{edit}^{'}(t) = \Delta_{ss} + \omega \cdot \Delta_{ts}^{orth} \quad (8)$$

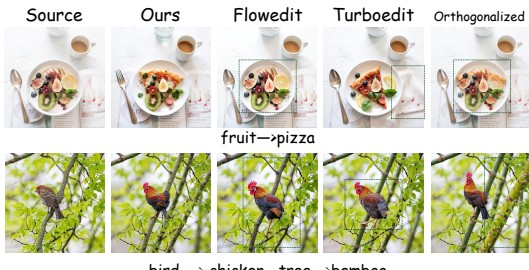

Figure 5: Qualitative comparison. Orthogonal calculation helps maintain background structure, and frequency domain control helps increase editing power.

### 3.3.2 FREQUENCY AWARE TRAJECTORY OPTIMIZATION

In multi-object editing scenarios, we found that simply amplifying the **cross-prompt** term does not effectively improve editing quality and can instead introduce unnecessary distortion, as shown in Fig. 5. By performing a frequency domain analysis of $V_{edit}$, we observed that the distribution of its high and low frequency components changes dynamically with the timestep. Therefore, we propose a **frequency-adaptive scaling strategy** to more precisely control the editing process.

Specifically, we first transform the editing vector into the frequency domain and apply adaptive scaling to its low- and high-frequency components. This process is defined as:

$$U_{\Delta t_i} = [\lambda_{low}(t_i) \cdot M_{low} + \lambda_{high}(t_i) \cdot M_{high}] \odot \mathcal{F}(\Delta t_i) \tag{9}$$

where $\mathcal{F}$ denotes the Fourier transform, and $M_{low}$ and $M_{high}$ are binary masks isolating the low and high frequency components, respectively. $\lambda_{low}(t_i)$ and $\lambda_{high}(t_i)$ are adaptive frequency scaling coefficients. The transformed vector is then converted back to the time domain:

$$\hat{V}_{\Delta t_i} = \mathcal{F}^{-1}(U_{\Delta t_i}) \tag{10}$$

The frequency scaling coefficient $\lambda_{type}(t_i)$ is computed based on the relative energy concentration within the residual spectrum $U_{\Delta t_i}$. This mechanism, controlled by parameter $\alpha$, compensates for missing frequency information and balances the contribution of low- and high-frequency components during image editing. The coefficient is defined as:

$$\lambda_{type}(t_i) = 1 + \alpha \left( 1 - \frac{\sum_{(k_x,k_y) \in R_{type}} |U_{t_i}(k_x, k_y)|}{\sum_{(k_x,k_y) \in R_{all}} |U_{t_i}(k_x, k_y)|} \right) \tag{11}$$

where $type \in \{low, high\}$, $R_{low}$ and $R_{high}$ denote the low and high frequency regions.

### 3.4 ATTENTION REMAPPING.

We construct the latent variable $Z_t$ by linearly interpolating between a clean image and Gaussian noise. However, as shown in the figure, a conflict between the source image's structure and the target prompt can lead to reconstruction errors such as **editing leakage**.

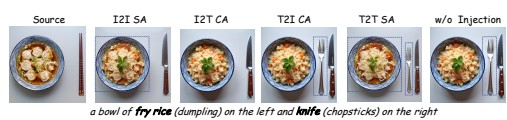

Source   I2I SA   I2T CA   T2I CA   T2T SA   w/o Injection

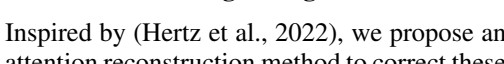

*a bowl of **fry rice** (dumpling) on the left and **knife** (chopsticks) on the right*

Figure 6: Edited results generated by injecting features. I2I-SA and I2T-CA can effectively preserve source image structure, I2I-SA interferes with subsequent text alignment.

Inspired by (Hertz et al., 2022), we propose an attention reconstruction method to correct these errors. The MMDiT joint attention module can be decoupled into four core components: I2I-SA, I2T-CA, T2I-CA, and T2T-SA. To evaluate their roles in image editing, we separately inject each component into the target branch. Our experiments showed that while both I2I-SA and I2T-CA can effectively preserve source image structure, I2I-SA interferes with subsequent text alignment. We therefore selected I2T-CA as the final choice.

For the $j$-th token in the target prompt, if a corresponding source token exists with an index of $\phi(j)$, we reuse the cross-attention value from the source as $B_{\phi(j)}$; otherwise, we amplify the original attention $A_j$ by a dynamic scaling factor $\beta$. Here, $A_j$ denotes the source's cross-attention (CA) value computed for the $j$-th token in the target prompt, and $B_{\phi(j)}$ is the corresponding attention value from the source prompt at index $\phi(j)$. The final attention fusion is formulated as:

$$A'_j = \omega_j \cdot B_{\phi(j)} + (1 - \omega_j) \cdot \beta \cdot A_j, \tag{12}$$

where $\omega_j = 1$ if a mapping exists, and $\omega_j = 0$ otherwise. The mapping function $\phi(j)$ assigns the index of the source prompt token corresponding to the $j$-th target token.

The amplification factor $\beta$ is dynamically adjusted based on the denoising timestep $t$:

$$\beta = \begin{cases} 1, & \text{if } t < T_{\text{turn}}, \\ \beta_0, & \text{if } t \geq T_{\text{turn}}. \end{cases} \tag{13}$$

This dynamic adjustment mechanism ensures that the source's CA prevents editing leakage or artifacts during the layout phase and the target's CA increases the editing intensity during the refinement phase. Furthermore, we found that each layer block of MMDiT also has a related frequency pattern. Please see the appendix for details. For both the layout and refinement phases, we select the corresponding attention layers to inject.

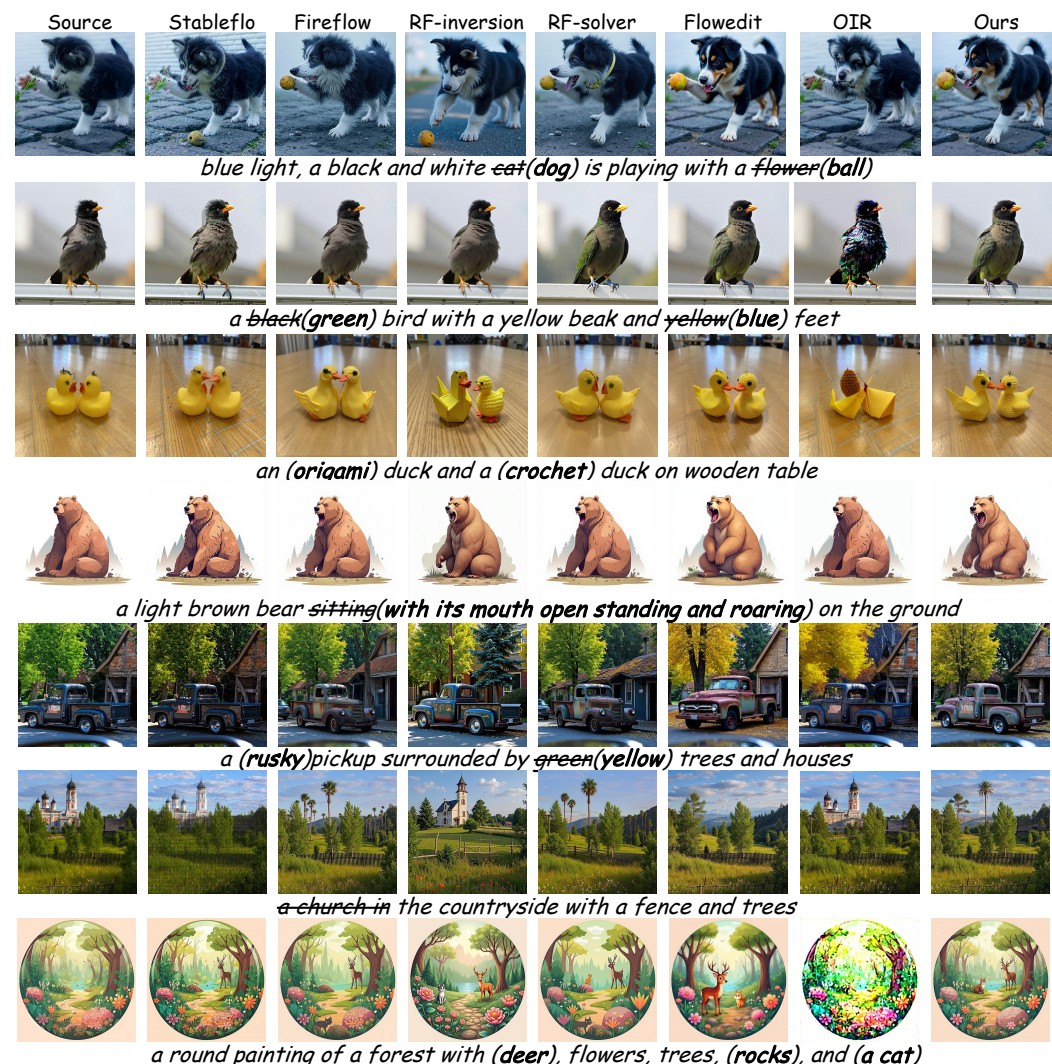

Figure 7: **Qualitative Comparison**. Unlike existing methods, our method allows users to add, replace, or modify multiple objects, making it highly efficient for editing complex scenes.

## 4 EXPERIMENT

### 4.1 IMPLEMENTATION DETAILS

#### 4.1.1 DATASETS.

For single-object editing, we evaluate our method and baseline methods on 9 tasks from PIE-Bench (Ju et al., 2023). For multi-object edit, we evaluate our method on three multi-object datasets: PIE-Bench++ (Huang et al., 2025), an augmented version of PIE-Bench designed for mixed edits involving 2-3 object categories; and the OIR (Yang et al., 2024) include mixed edits across two different task types. All three datasets provide paired sources and target prompts.

#### 4.1.2 METRICS.

We evaluated our method from two perspectives: (a) source preservation and (b) text alignment. For source preservation, we measured Structure Distance Tumanyan et al. (2022a), PSNR Huynh-Thu and Ghanbari (2008), LPIPS Zhang et al. (2018), MSE, and SSIM. Note that the numbers of these

metrics reported in this paper are scaled. For text alignment, we measure the CLIP similarity Radford et al. (2021) between the whole image and the target prompt (Whole) and

### 4.1.3 BASELINES AND IMPLEMENTATION DETAILS

We mainly compare our method with previous state-of-the-art training-free image editing methods. For RF-based models, we evaluate RF-Inversion (Rout et al., 2024), RF-solver (Wang et al., 2025), FireFlow (Deng et al., 2024), Stable Flow (Avrahami et al., 2025), and FlowEdit (Kulikov et al., 2025). For DMs, we evaluated DDIM+P2P (Hertz et al., 2022), Direct Inversion+PnP (Tumanyan et al., 2022b), Infedit (Xu et al., 2023), MasaCtrl (Cao et al., 2023). In addition, we also include multi-object editing method OIR (Yang et al., 2024). We follow their official implementations for evaluation. We implement our method based on FLUX (Labs, 2024). Throughout the comparative evaluations, our hyperparameters remain fixed: $\beta = 4$ for cross attention injection. Specifically, during the Layout Phase, cross attention from the source image is injected into layers 5-20 of the target branch. During the Refinement Phase, the injection occurs in layers 20-45. Further implementation details are provided in Appendix.

### 4.2 EDITING RESULTS

#### 4.2.1 QUALITATIVE EVALUATION.

Qualitative results are shown in Fig. 7. From our experiments, we observe the following: Firstly, methods such as StableFlow, Fireflow, and RF-inversion often exhibit omitted edits when dealing with complex scenes involving multiple materials, colors, or object modifications, leading to noticeable text-image misalignment. Secondly, for multi-object editing approaches like OIR , their performance is suboptimal in non-rigid editing tasks, such as object addition or removal. This is primarily because these methods heavily rely on precise masks for editing, which makes it challenging to generate natural and contextually consistent results when an object needs to be completely removed or created from scratch. Thirdly, FlowEdit demonstrates better editing performance in certain scenarios; however, it overlooks the variations in inversion steps across different images, which can lead to suboptimal editing outcomes.

| Method | Structure | Background Preservation | | | | CLIP Similarity | |
| | Distance $\times 10^3$ ↓ | PSNR ↑ | LPIPS $\times 10^2$ ↓ | MSE $\times 10^3$ ↓ | SSIM $\times 10^2$ ↑ | Whole ↑ | Edited ↑ |
|---|---|---|---|---|---|---|---|
| DDIM+P2P | 69.43 | 17.87 | 20.88 | 21.99 | 71.14 | 25.01 | 22.44 |
| DI+PnP | 24.29 | 22.46 | 10.61 | 8.045 | 79.68 | 25.41 | **22.62** |
| InfEdit | 13.78 | **28.51** | 4.758 | 3.209 | 85.66 | 25.03 | 22.22 |
| MasaCtrl | 28.07 | 22.18 | 10.15 | 8.677 | 80.26 | 24.96 | 21.40 |
| RF-Inversion | 32.62 | 22.03 | 15.96 | 9.601 | 73.26 | 24.89 | 21.89 |
| RF-Solver | 24.17 | 26.12 | 11.88 | 4.064 | 86.50 | 25.19 | 22.07 |
| StableFlow | 14.41 | 25.98 | 7.246 | 4.471 | 92.08 | 24.20 | 20.86 |
| FireFlow | 22.42 | 25.91 | 11.45 | 4.396 | 86.56 | 25.41 | 22.08 |
| FlowEdit | 21.07 | 23.59 | 8.889 | 6.631 | 88.89 | 24.90 | 21.66 |
| Ours | **8.754** | 28.50 | **4.143** | **2.111** | **94.69** | **25.44** | 22.00 |

Table 1: Comparison with different baselines for single-object edits in PIE benchmark. The best score is highlighted in **bold**, and the second-best score is underlined.

#### 4.2.2 QUANTITATIVE EVALUATION.

For single-object edits, the quantitative results are summarized in Table 1. Our method performs well on most metrics, with the exception of the edited CLIP score and PSNR. Although DI+PnP obtains a higher CLIP score, its ability to preserve the background and structure of the source image is inferior to that of our method. For multi-object edits, the quantitative results are summarized in Table 2. Our method achieves good performance on most metrics. Specifically, while InfEdit shows better PSNR, its CLIP similarity is significantly lower than other methods. This suggests that its strong background preservation capability hinders its editing ability. FlowEdit achieves a high CLIP score but shows weak background and structure preservation. In summary, our method performs well in both background preservation and editing, enabling precise editing without compromising structural consistency or editability. Quantitative results for OIR are shown in the Appendix.

| Method | Structure | Background Preservation | | | | CLIP Similarity | |
|---|---|---|---|---|---|---|---|
| | Distance $\times 10^3$ ↓ | PSNR ↑ | LPIPS $\times 10^2$ ↓ | MSE $\times 10^3$ ↓ | SSIM $\times 10^2$ ↑ | Whole ↑ | Edited ↑ |
| DDIM+P2P | 43.57 | 18.48 | 18.83 | 19.01 | 73.55 | 20.72 | 19.58 |
| DI+PnP | 27.07 | 22.73 | 10.32 | 7.597 | 80.73 | 20.79 | 19.07 |
| InfEdit | 22.60 | 24.61 | 10.40 | 16.05 | 78.85 | 24.69 | 22.63 |
| MasaCtrl | 29.76 | 22.50 | 10.22 | 8.758 | 81.59 | 24.15 | 22.14 |
| RF-Inversion | 42.29 | 21.41 | 18.19 | 11.24 | 73.70 | 25.06 | 23.29 |
| RF-Solver | 23.83 | 26.65 | 11.523 | 3.778 | 87.26 | 24.46 | 22.80 |
| StableFlow | 16.43 | 26.38 | 6.582 | 4.038 | 91.76 | 22.79 | 21.48 |
| FireFlow | 20.49 | 27.06 | 10.50 | 3.854 | 88.21 | 24.52 | 22.80 |
| FlowEdit | 22.52 | 24.00 | 8.638 | 6.206 | 89.78 | 25.11 | 23.43 |
| OIR | 24.66 | 27.79 | **5.715** | 2.460 | 86.97 | 23.56 | 20.95 |
| Ours | **12.82** | 27.84 | 6.523 | **2.430** | **91.83** | **25.65** | **23.55** |

Table 2: Comparison with different baselines for multi-object edits in PIEBench++. The best score is highlighted in **bold**, and the second-best score is underlined.

### 4.3 ABLATION STUDY

We conducted an ablation study on three core technical components of our method: starting point optimization, feature injection, and difference modulation. Table 3 presents quantitative results. Without starting point optimization, a significant portion of the source structure is lost in the edited images. This may stem from extracting features from the latent space that contain inaccurate information about the source image. Meanwhile, without feature injection, the edits fail to fully align with the target prompt, leading to a lower text-alignment score. Furthermore, without trajectory optimization(TO), edits maintain high image similarity but yield lower CLIP scores, potentially due to insufficient editing strength. These findings collectively underscore the importance of

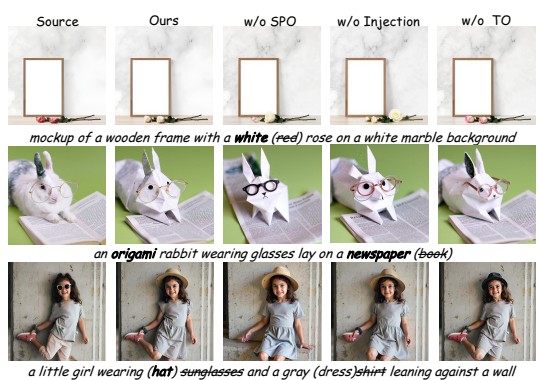

Figure 8: Ablation examples for assessing the impact of each technique in our method.

both source structure preservation and semantic alignment in image editing, objectives that our method effectively achieves. Fig. 8 provides a qualitative comparison. As long as the correct starting point is determined, the original image structure can be maintained. The other two strategies will improve the text alignment.

| Method | Structure | Background Preservation | | | | CLIP Similarity | |
|---|---|---|---|---|---|---|---|
| | Distance $\times 10^3$ ↓ | PSNR ↑ | LPIPS $\times 10^3$ ↓ | MSE $\times 10^3$ ↓ | SSIM $\times 10^2$ ↑ | Whole ↑ | Edited ↑ |
| w/o SPO | 17.26 | 25.82 | 8.072 | 4.057 | 90.10 | 23.78 | 22.28 |
| w/o Injection | 15.00 | 26.14 | 7.749 | 3.374 | 90.11 | 24.54 | 22.70 |
| w/o Modulation | 13.45 | 27.72 | 6.256 | 2.650 | 91.06 | 24.24 | 22.65 |
| Ours | **12.82** | **27.84** | **5.523** | **2.430** | **91.83** | **25.65** | **23.55** |

Table 3: **Ablations Study**. Three strategies com plement each other and result in improved metrics.

## 5 CONCLUSION

In conclusion, we analyze the intermediate features at each timestep of inversion-free methods, and propose a simple yet effective starting point optimization strategy. In addition, we introduce Trajectory Optimization method to address the editing omission and detail loss problems in multi-object and complex scene editing. We hope that our analysis of start point and frequency will serve as a building block for future advancements in image editing.

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

# A APPENDIX

## THE USE OF LLM

We acknowledge the use of a large language model (ChatGPT, GPT-5 by OpenAI) to assist in improving the clarity and readability of the manuscript. The model was used only for language polishing (e.g., grammar checking, wording refinement) and not for generating novel scientific content, experimental design, data analysis, or results. All technical ideas, methods, and contributions are solely the work of the authors.

## A. DETAILED ANALYSIS

### A.1 START POINT

As shown in Fig. 9, we present three examples of images with different levels of complexity. For a 50-step process, their starting point is approximately the 10th timestep.

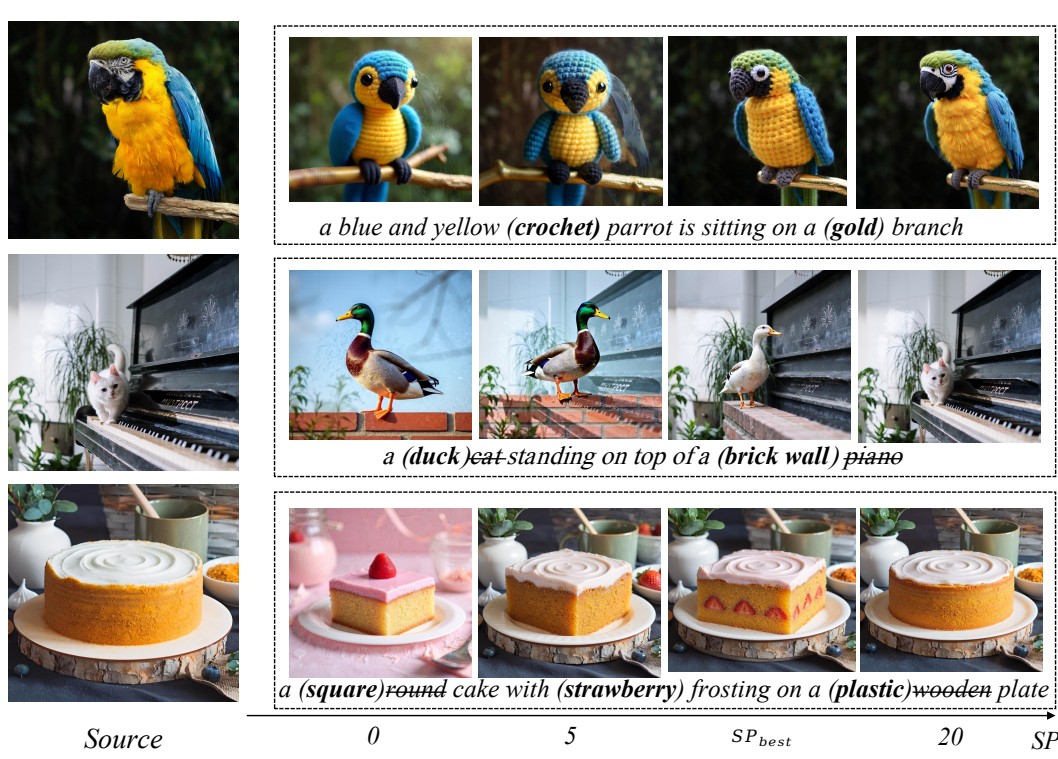

Figure 9: **Analysis of start point**

### A.2 ATTENTION LAYERS

As shown in Fig. 10, The MM-DiT architecture exhibits consistent frequency-domain characteristics across different timesteps. The model, which consists of 57 blocks, processes information in a hierarchical manner. As the layer number increases, the attention outputs progressively incorporate higher frequency components. Specifically, the early layers establish the low frequency structure of the image, such as object placement and motion, while the later blocks focus on high frequency details. Consequently, to inject low frequency structural information, we should concentrate on the early layers. Conversely, for injecting high frequency details, attention should be directed to the later layers.

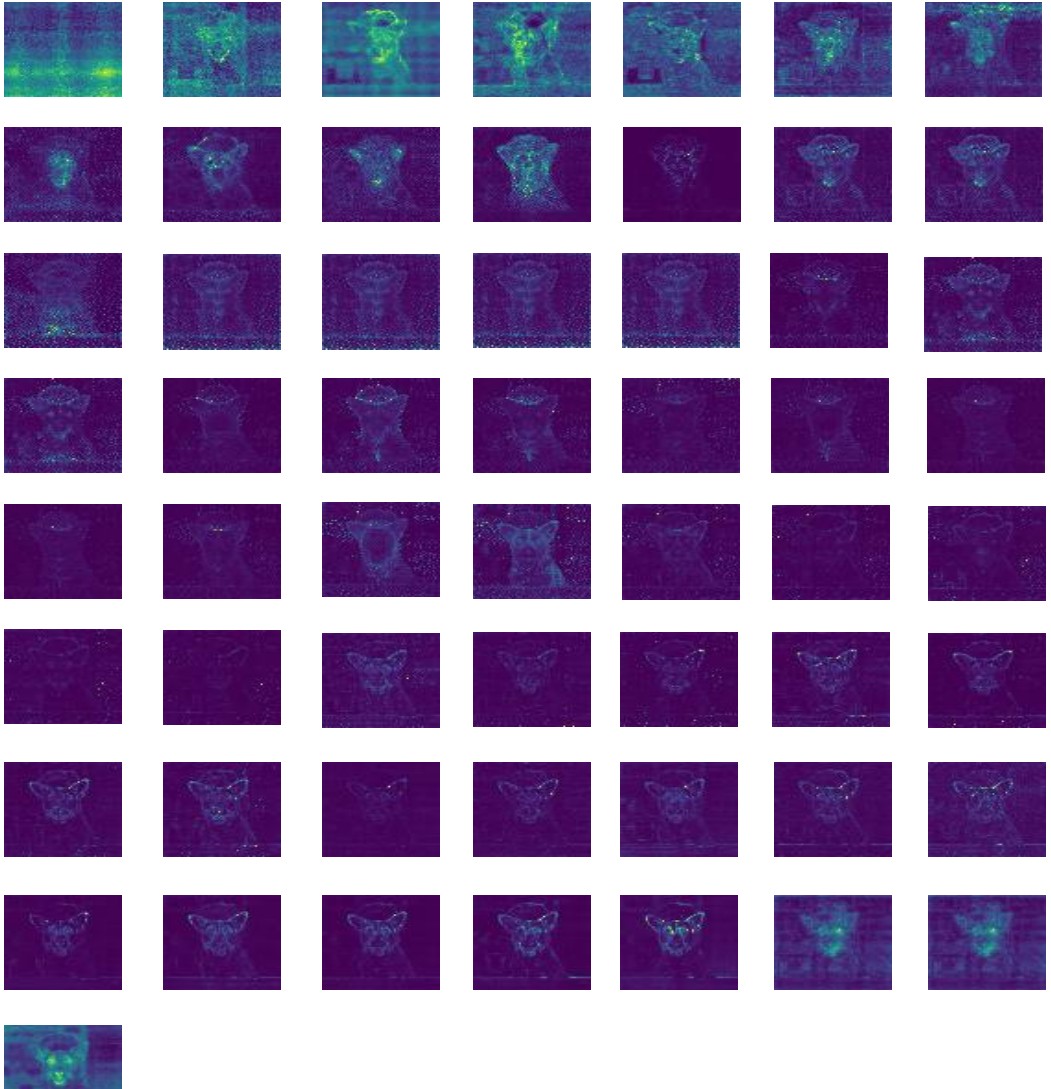

Figure 10: **Visual analysis of the layers of MM-DiT at each timesteps**.

## B. IMPLEMENTATION DETAILS

### B.1 INJECTION LAYERS

The MM-DiT architecture exhibits similar frequency-domain properties. To leverage this, we conducted ablation studies to determine the optimal attention injection strategy for both the **layout phase** and the **refinement phase**.

As shown in Fig. 10, we observed that the first five layers contain relatively limited information. For the layout phase, we set the injection starting point at layer 5. As can be seen from Fig. 11. Injecting source image attention from layers 5 to 20 effectively removes editing artifacts. It is noteworthy that a narrower injection range, such as layers 5-10 or 5-15, tends to generate unrelated objects. Conversely, extending the range to layer 25 excessively preserves source details, which constrains editing flexibility. Furthermore, if the injection starts later at layer 10, editing artifacts reappear, proving that layers 5-10 are indispensable.

For the refinement phase as shown in Fig. 12, we found that injecting attention from the 20th block effectively enhances editing strength. This strategy aims to prevent editing omissions because the first 20 blocks are primarily responsible for the image's macro layout. Experiments indicate that layers 20-45 are the optimal injection range. Using layers 20-35 still leads to editing leakage, while extending the range to 20-55 significantly limits the editing effect. Additionally, we found that injecting only into layers 5-20 does not improve performance, and the results from layers 5-45 are similar to those from layers 20-45.

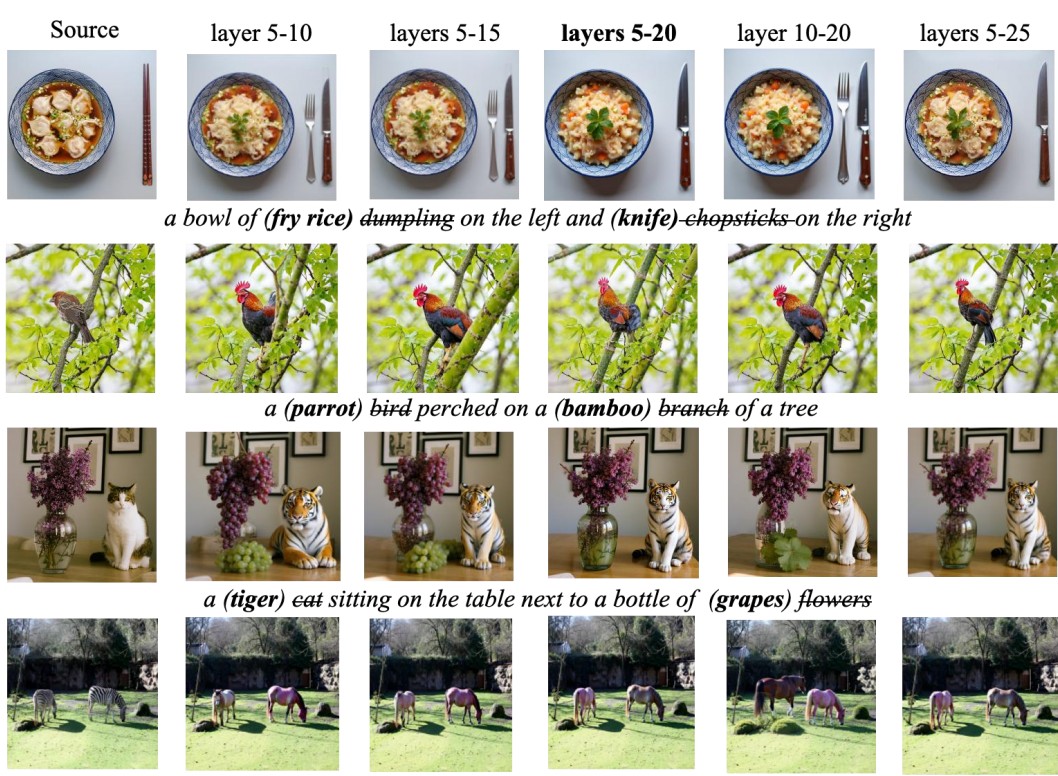

Figure 11: **Analysis of injection layers in layout phase**

### B.2 BASELINE IMPLEMENTATION

In this section, we describe the implementation details of the baselines we used.

For **RF-Inversion**, we follow their Github official implementation, the stopping timestep is set to 7/28, and the strength is set to 0.9.

For **RF-Solver-Edit**, we follow their Github official implementation, guidance is set to 2 and inject is set to 5.

For **Stable Flow**, we follow their Github official implementation, the vital layers are the same as the official implementation.

For **FireFlow**, we follow their Github official implementation, the number of steps is set to 8, and the inject is set to 1.

For **FlowEdit**, we follow their Github official implementation, and use their default setting for FLUX.

For **OIR**, we follow their Github official implementation, reinversion step and reassembly step are both set to 10.

For **InfEdit**, we follow their Github official implementation.

For **DDIM+P2P**, **DI+PnP** and **MasaCtrl**, we follow the implementations from **Direct Inversion** codebase

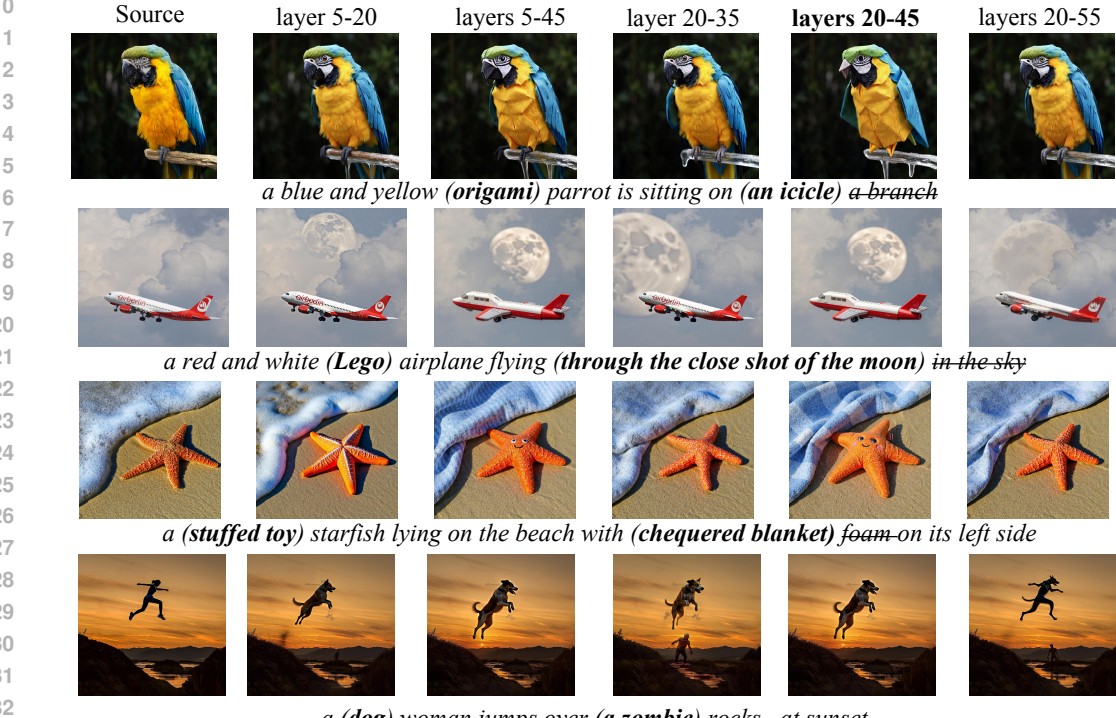

| Source | layer 5-20 | layers 5-45 | layer 20-35 | **layers 20-45** | layers 20-55 |

*a blue and yellow (**origami**) parrot is sitting on (**an icicle**) a branch*

*a red and white (**Lego**) airplane flying (**through the close shot of the moon**) in the sky*

*a (**stuffed toy**) starfish lying on the beach with (**chequered blanket**) foam on its left side*

*a (**dog**) woman jumps over (**a zombie**) rocks at sunset*

Figure 12: **Analysis of injection layers in refinement phase**

## C. ADDITIONAL QUALITATIVE EVALUATION

We performed additional qualitative comparisons against baselines. Extended comparison results on single-object and multi-object are presented in Fig. 13 and Fig. 14, with more examples in Fig. 15 and Fig. 16.

## D. ADDITIONAL QUANTITATIVE COMPARISON

We present the comprehensive results of our quantitative evaluation on the OIR-Bench and LoMOE-Bench dataset in Tab 5 and Tab **??**. To assess how well the original image's structure is preserved, we used several metrics: Structure Distance, PSNR, LPIPS, MSE, and SSIM. To measure how closely the edited image aligns with the text prompt, we computed CLIP text similarity. This was done in two ways: one for the entire image and another specifically for the editing mask region, which we refer to as "Whole Image Clip Similarity" and "Edit Region Clip Similarity."

### D.1 OIR-BENCH

Our method achieves better perfomance across most metrics. Concretely, the results show that LoMOE has a stronger ability to preserve source content, but this also limits its editing flexibility. As shown in the CLIP scores, LoMOE performs relatively worse in terms of edited similarity. In summary, our method performs well in both background preservation and editing, enabling accurate edits without compromising structural consistency or editability.

### D.2 DETAILS ON USER STUDY

We conducted two user studies, comparing our method against eight multi-object editing techniques, and eight single-object editing methods. For the studies, we selected 8 images each from single-object and multi-object editing tasks within our dataset. These images were generated from the same source image and target prompt but using different methods. More than 20 participants are asked to select the image that best conformed to the target prompt while effectively preserving the source image structure. As show in Table 4. We carried out a user study to compare image editing methods, involving 28 anonymous Prolific users and 14 questions. Participants were shown a source image and an editing instruction. Their task was to select one of eight edited images based

on two criteria: edit accuracy and backgroud preservation . This process, exemplified in Fig. 17, allows us to analyze user preferences and inform the development of more effective and precise editing methods.

| Comparison on single-object edits | | | | | | | |
|---|---|---|---|---|---|---|---|
| Method | DI+PnP | MasaCtrl | Infedit | RF-Inversion | RF-Edit | FlowEdit | Stableflow | **Ours** |
| User Preference | 4.3% | 3.4% | 8.4% | 6.7% | 7.3% | 9.1% | 10.3% | **50.5%** |

| Comparison on multi-object edits | | | | | | | |
|---|---|---|---|---|---|---|---|
| Method | DI+PnP | Inf-Edit | RF-Inversion | RF-Edit | FlowEdit | Stableflow | OIR | **Ours** |
| User Preference | 5.6% | 3.2% | 6.8% | 7.2% | 9.8% | 7.2% | 5.4% | **54.8%** |

Table 4: User study results comparing our method with nine methods in single-object edits and multi-object edits.

# E. ADDITIONAL EXAMPLES OF ABLATION STUDIES ON EACH TECHNIQUE

## E.1 EFFECT OF VARYING $\beta$ IN CROSS-ATTENTION INJECTION

The effect of the hyperparameter $\beta$ is presented in Fig. 19. With $\beta=1$, the model fails to fully align the edited image with the text prompt. Increasing $\beta$ significantly improves this alignment, as indicated by better comparison scores. However, an excessively large $\beta$ can't increase the editing effect Therefore, selecting an appropriate $\beta$ is crucial. Based on this observation, we set $\beta=4$ for all subsequent experiments.

## E.2 EFFECT OF VARYING START POINT

The effect of the hyperparameter start point is presented in Fig. 18. Starting editing too early will destroy the original image structure, and starting editing too late will not align with the text prompt.

# F. FUTURE WORK

We believe that image processing and video processing are inseparable from the frequency domain. Later we will continue to explore the role of the frequency domain in the image and video fields.

| | Structure | Background Preservation | | | | CLIP Similarity | |
|---|---|---|---|---|---|---|---|
| **Method** | Distance $\times 10^3$ ↓ | PSNR ↑ | LPIPS $\times 10^2$ ↓ | MSE $\times 10^3$ ↓ | SSIM $\times 10^2$ ↑ | Whole ↑ | Edited ↑ |
| DI+PnP | 33.74 | 22.60 | 10.88 | 6.796 | 83.30 | 28.18 | 25.56 |
| MasaCtrl | 26.62 | 22.76 | 10.67 | 7.261 | 81.44 | 21.81 | 19.79 |
| RF-Inversion | 44.89 | 21.01 | 16.56 | 9.95 | 79.81 | 27.72 | 25.32 |
| RF-Solver | 27.02 | 25.17 | 10.39 | 4.25 | 87.45 | 27.23 | 25.14 |
| StableFlow | **16.03** | 24.46 | 7.28 | 5.23 | 91.61 | 22.51 | 20.29 |
| FireFlow | 26.84 | 24.44 | 11.28 | 4.65 | 86.10 | 26.95 | 24.26 |
| FlowEdit | 31.59 | 22.81 | 9.256 | 6.92 | 88.43 | 27.07 | 24.70 |
| OIR | 21.56 | 28.65 | 4.608 | 2.342 | 88.11 | 28.74 | **26.51** |
| Ours | 18.47 | **30.53** | **3.295** | 2.910 | **92.27** | **28.88** | 26.09 |

Table 5: Comparison with different baselines for multi-object edits in OIR Bench. The best score is in **bold**, and the second-best score is underlined.

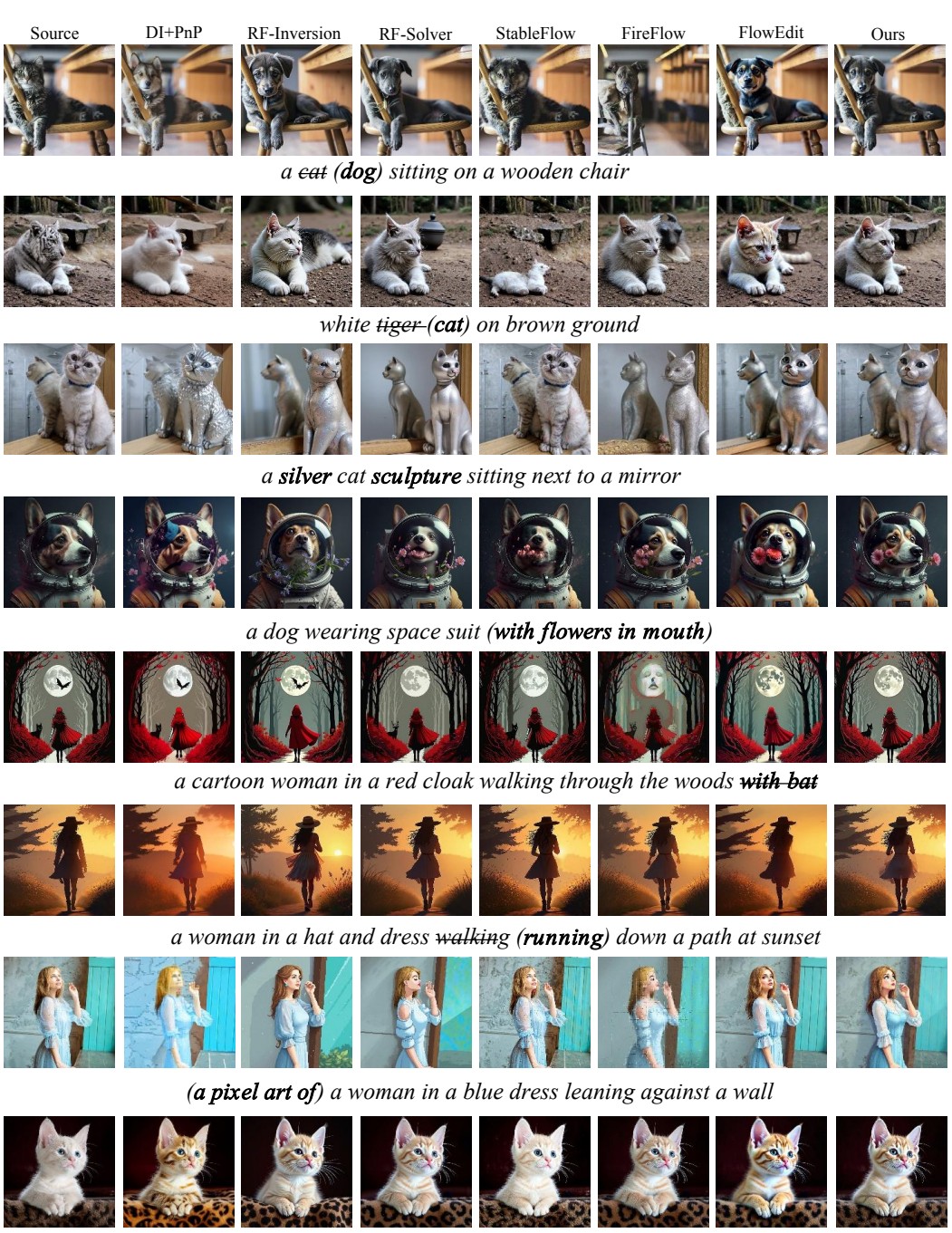

Figure 13: Additional evaluation comparisons on single-object editing.

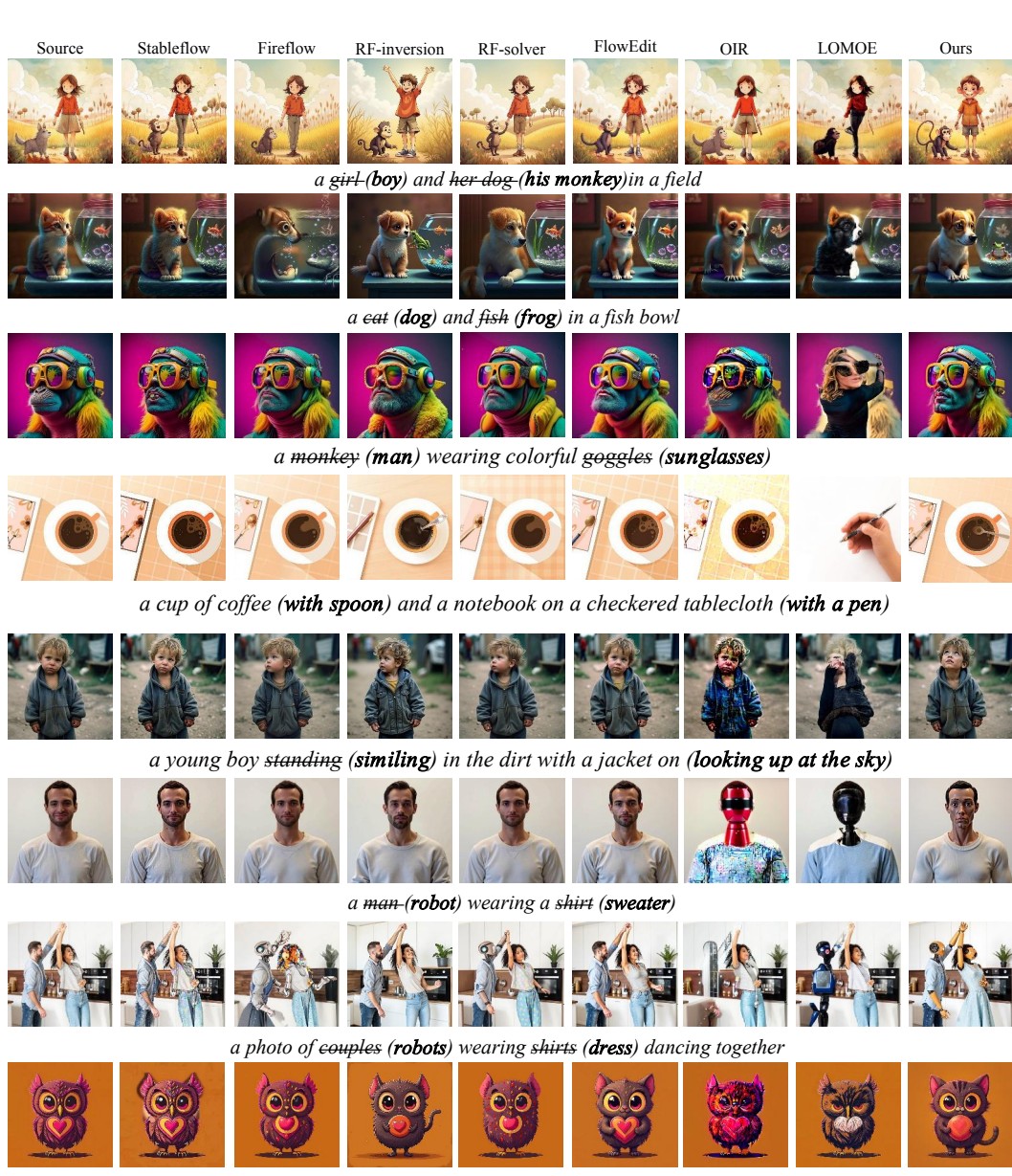

Figure 14: Additional evaluation comparisons on multi-object editing.

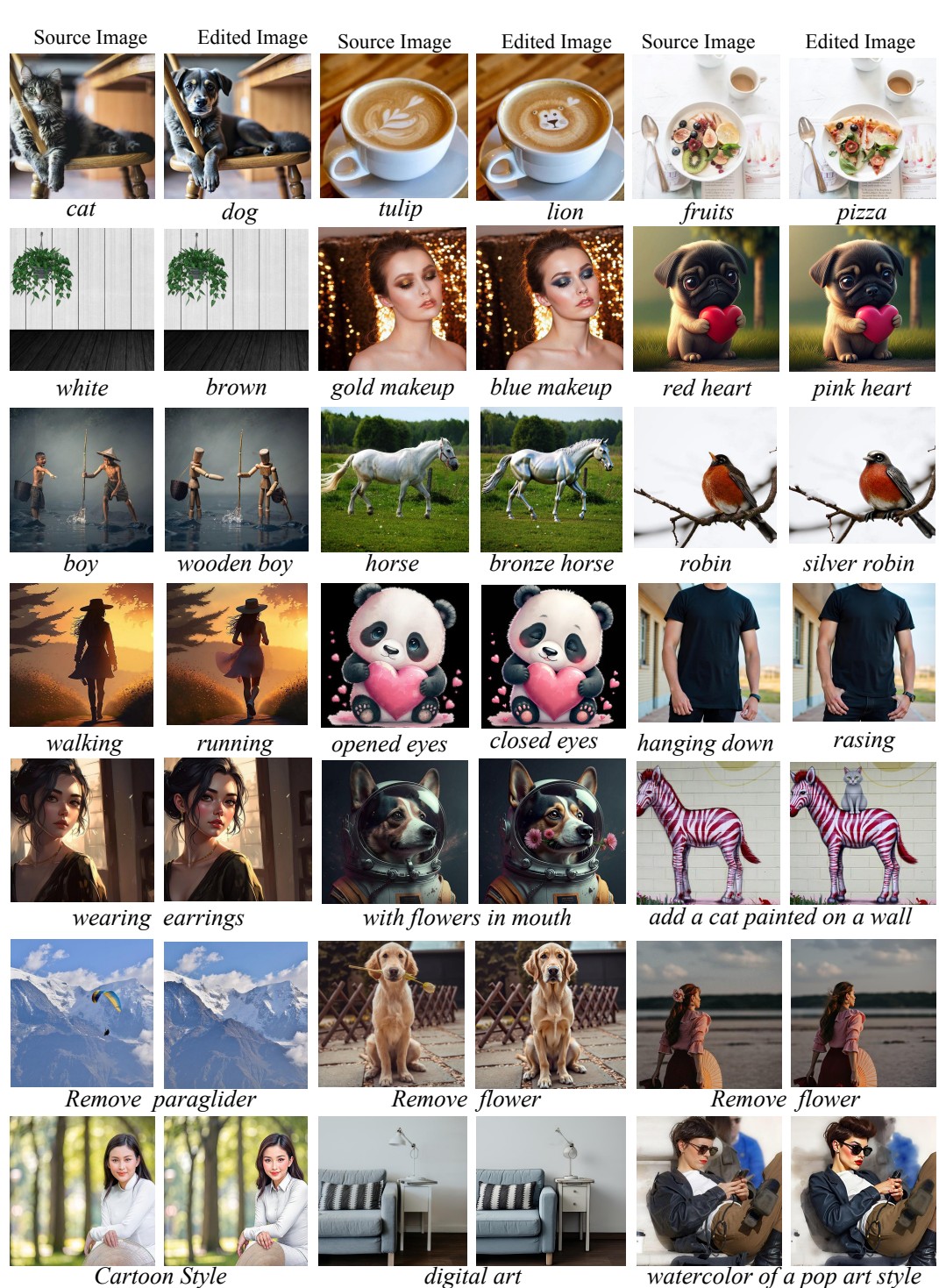

Figure 15: **Diverse edited results of our method on single-object editing**. Our method allows users to add, replace, change object, change color and change material.

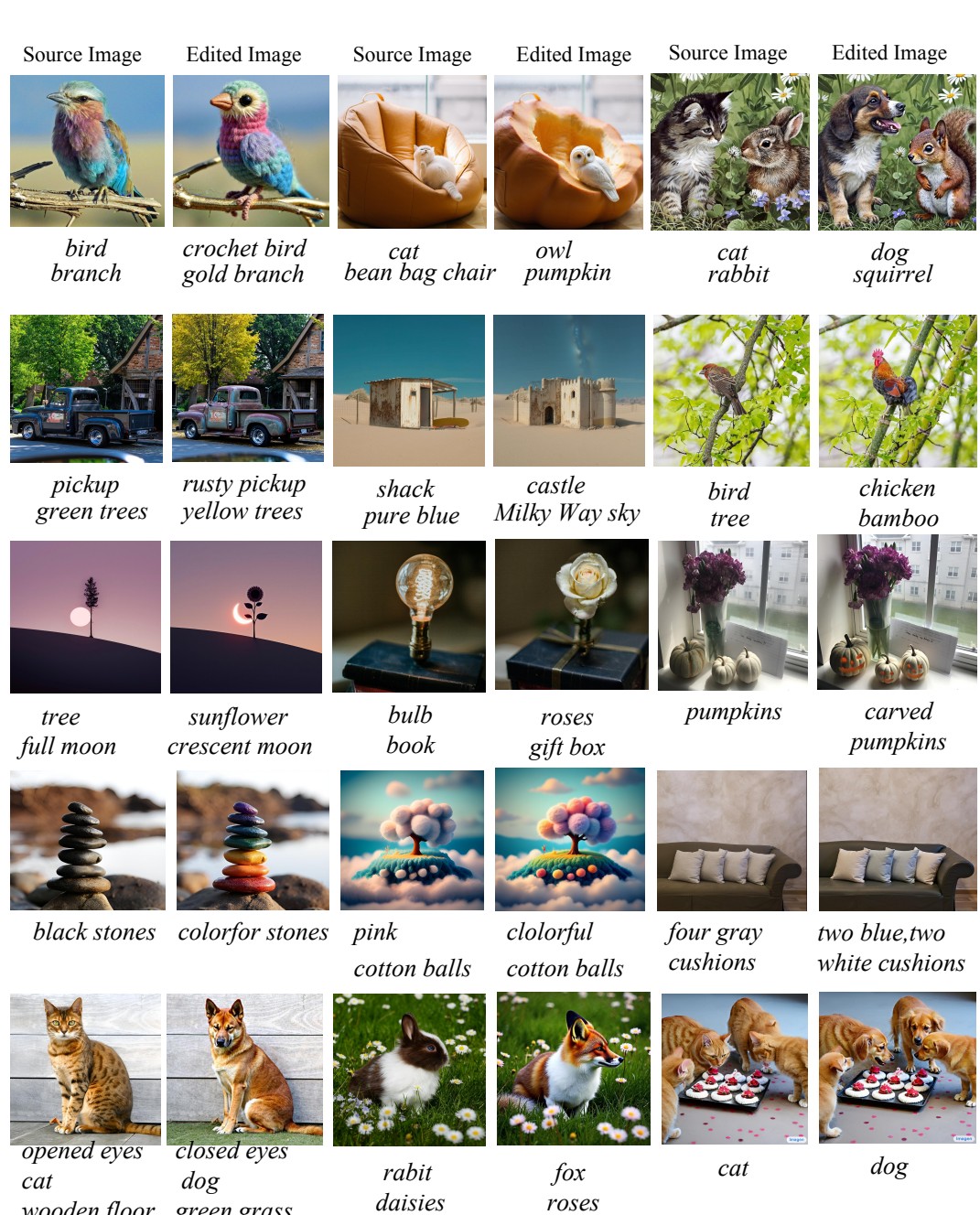

Figure 16: **Diverse edited results of our method on multi-object editing**. Our method allows users to add, replace, change object, change color and change material.

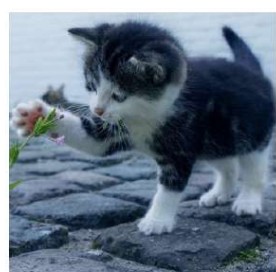

**Original prompt:** blue light, a black and white cat is playing with a flower

**Edit prompt:** blue light, a black and white **dog** is playing with a **yellow ball**

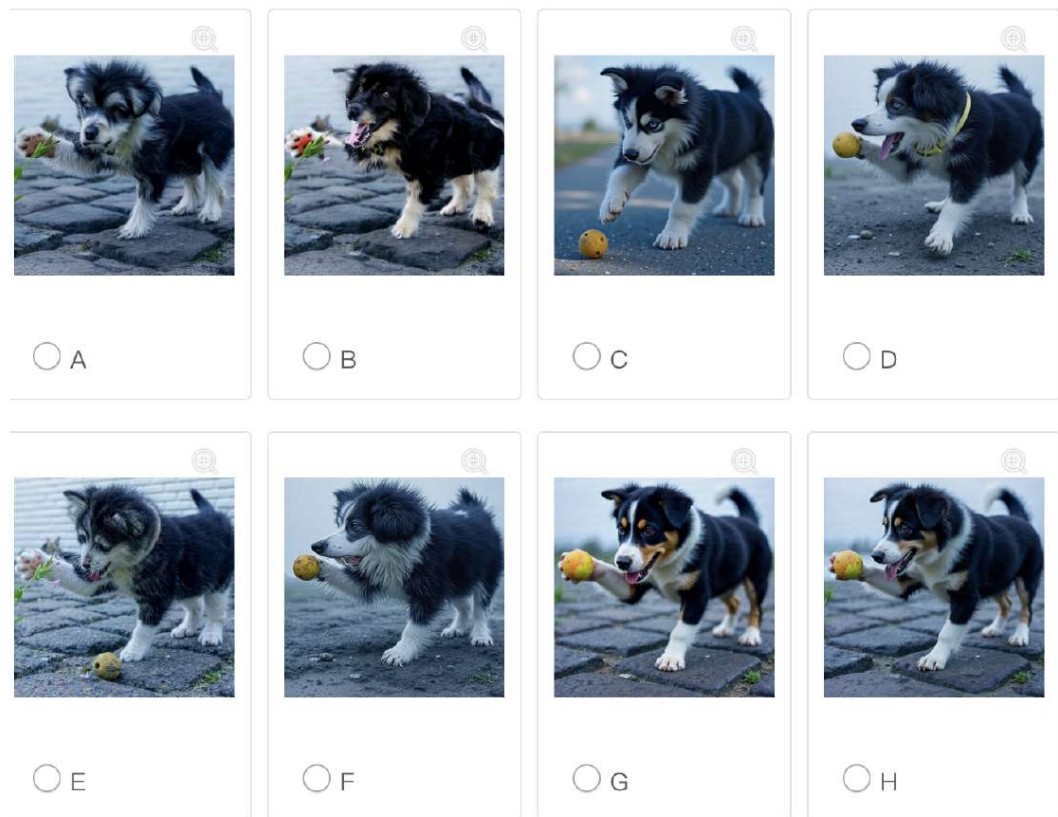

Figure 17: Example screenshot from the user study, displaying images generated using different methods, where participants selected the one that best represents the intended edit.

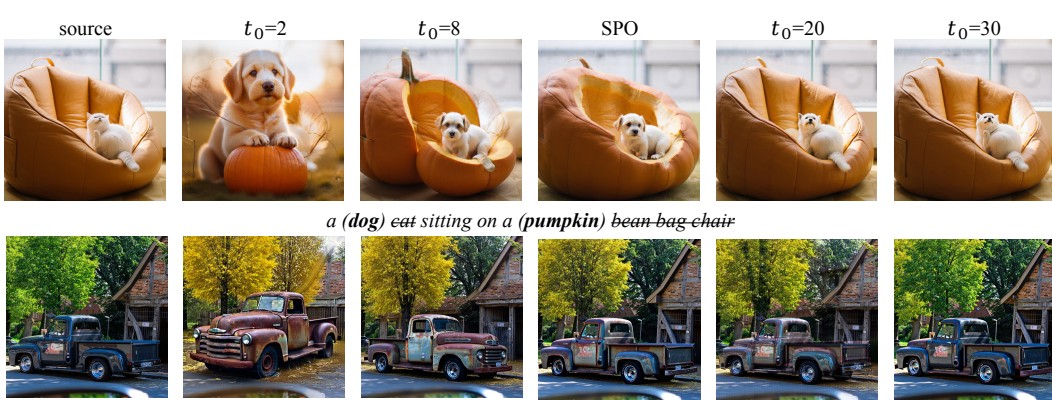

Figure 18: The effect of start point

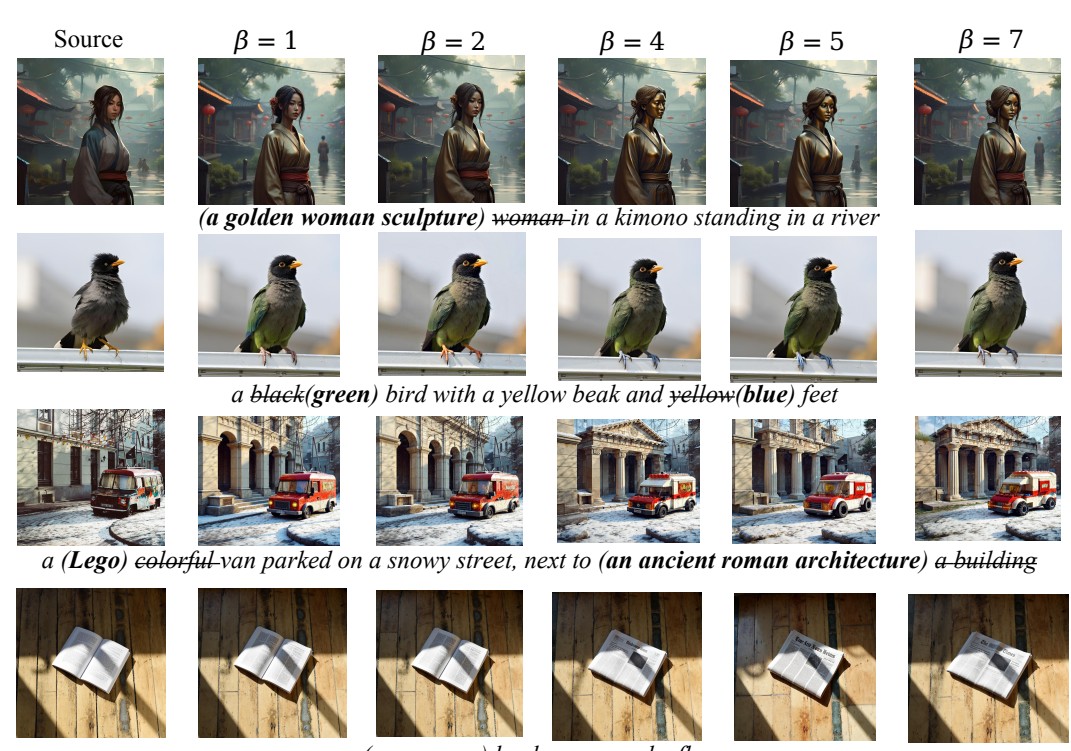

Figure 19: The effect of $\beta$

