# OpenReview forum: "Flowing with Precision: Rectified Flow Image Editing with Trajectory and Frequency Guidance"
_ICLR.cc/2026/Conference — ICLR 2026 Conference Withdrawn Submission_

### Official Review · Reviewer_h4Yd · 2025-10-24

**Soundness:** 2
**Presentation:** 3
**Contribution:** 2
**Rating:** 4
**Confidence:** 4

**Summary:**

The paper proposes a rectified-flow-based image editing framework that improves semantic alignment and structural preservation in complex, multi-object scenes. The key innovations include:
1. Starting Point Optimization (SPO) – adaptively choosing the optimal timestep to start editing.
2. Trajectory Optimization (TO) – orthogonalizing semantic and structural editing vectors in both time and frequency domains.
3. Attention Remapping – selective feature injection across MM-DiT layers for layout and refinement phases.
4. Experiments on PIEBench, PIEBench++, and OIR-Bench demonstrate superior fidelity and flexibility over RF-based and diffusion-based baselines.

**Strengths:**

1. Clear problem definition and systematic analysis of editing phases.
2. Well-motivated dual-domain (time + frequency) optimization.
3. Impressive visual results and detailed ablation/user studies.
4. Training-free, plug-and-play design suitable for practical use.

**Weaknesses:**

1. Limited theoretical novelty.
SPO and frequency-aware scaling are clever extensions but largely heuristic; they build upon existing inversion-free frameworks like FlowEdit and FlowAlign with incremental innovation. The paper could more explicitly differentiate its contributions from prior flow-guided works.

2. Complexity and interpretability.
The method involves several intertwined components (SPO, vector orthogonalization, FFT modulation, multi-phase attention). It is unclear which components most critically drive improvements—despite ablations, the causal links are not deeply analyzed.

3. Computational efficiency.
The approach adds frequency transforms and per-timestep adaptive scaling, potentially increasing runtime. There is no comparison of inference time or memory usage against baselines such as FireFlow or StableFlow.

4. Heuristic parameter tuning.
The determination of SPO start point (via low-frequency MSE) and scaling factors (α, β) are empirically set. The paper lacks sensitivity analysis or robustness evaluation for these hyperparameters.

5. Scope of generalization.
While results on multi-object scenes are impressive, the paper focuses only on static images. Discussion of extension to video or 3D scenes would improve its broader impact.

**Questions:**

1. How sensitive is performance to the SPO threshold or to the frequency weighting parameter α?
2. What is the computational overhead compared to FlowEdit or StableFlow?
3. Could the same idea be generalized to temporal consistency in video editing?

---

### Official Review · Reviewer_9Q8V · 2025-10-31

**Soundness:** 3
**Presentation:** 1
**Contribution:** 2
**Rating:** 2
**Confidence:** 5

**Summary:**

This work proposes an integrated editing framework, leveraging three distinct components targeting at different aspects of diffusion model's generation process to perform image edting. The framework combines several technical operations, including identification of different stages of diffusion generation process, semantic isolation by orthogonal projection, frequency modulation and attention injection. All these techniques reflect the up-to-date progress of image editing with diffusion models comprehensively. With such effort the proposed editing framework achieves good editing results on both PIE and OIE benchmarks.

**Strengths:**

The paper tackles the image editing task from different aspects of diffusion models,  including:
-  The generation machnism like different stages of generation process and frequency evolving characterstics, that is being studied to network components (attention modules).
- The semantic space of diffusion models and corresponding operations.
- The widely studied and applied attention operation to fuse generation from two branches/paths.

Specifically, the paper proposes:

1. Identifying different stages of generation process via comparing the similarity of scores(diffusion network output) from different branches.
2. Semantic isolation of the cross branch to decrease the constraint from the source structure.
3. Frequency bands modulation on the overall editing score $V_{edit}$ to eliminate distortion.
4. Attention injection to prevent editing leakage.

With combining the technical components together, the proposed editing framework achieve good editing results on two general editing benchmarks.

**Weaknesses:**

1. Several of the proposed technical components lack clear motivation or demonstration of originality. For instance, regarding the orthogonal projection of the cross-prompt vector onto the cross-trajectory vector, it remains unclear how this improvement was discovered or why the projection is implemented in the current manner rather than an alternative formulation. Similarly, for the frequency-modulation component, the paper does not explain how the authors identified this particular frequency as crucial for addressing distortion, nor why Eq. (11) provides an effective solution to this issue. **The absence of explicit motivation and analytical justification for these technical designs hinders the perceived originality of the work, especially given that similar components have appeared in prior studies.**

2. Important implementation details and hyperparameters are missing. For example, the numerical value of $\alpha$ used in frequency modulation is neither specified nor discussed in the main text or the supplementary materials. Although the authors mention setting $\beta = 4$, the initial value $\beta_0$ is not clarified. Additionally, the definition of the boundary between high- and low-frequency regions is omitted. Other key parameters used during the editing process, such as $T_{\text{turn}}$, are also unspecified — it is unclear whether this is fixed to a common value such as $T_0$ or determined individually for each editing pair.

3. Most proposed components are insufficiently analyzed through experiments. The ablation study only reports quantitative benchmark results obtained by removing individual components, without providing deeper empirical investigations into the choice or sensitivity of corresponding hyperparameters.

4. The entire editing framework integrates multiple components, temporal operations across different diffusion timesteps, and layer-specific attention injections. To improve clarity and reproducibility, the paper should include a pseudo-code algorithm summarizing the complete framework.

**Questions:**

Please check the weakness mentioned above. And below are some further suggestions:

(1) The Eq.(6) should be moved to preliminary since it is proposed in TurboEdit and serves as one of their main contributions.

(2) There is missing space for $T_{0}$ $T_{turn}$ in Sec 3.2

(3) Please add clear definition to $V^{src}_{tar}$, $V^{src}_{src} $, $V^{tar}_{tar}$, in Fig.4 to avoid confusion.

Rating for this paper would be impoved if the questions are addressed.

**Details Of Ethics Concerns:**

None.

---

### Official Review · Reviewer_TexM · 2025-10-31

**Soundness:** 3
**Presentation:** 2
**Contribution:** 3
**Rating:** 2
**Confidence:** 4

**Summary:**

The paper introduces a training-free image editing method based on text-to-image flow models. The method is divided into two main parts. First, a module called Starting Point Optimization that determines the “optimal” editing starting timestep according to the spectral components of the intermediate edit directions. Second, a module called Trajectory Optimization which decomposes the commonly-used editing direction into a cross-prompt term and a cross-trajectory term, and forces orthogonalization between them. Additionally, this module employs frequency domain reweighting, and uses attention feature injection. The authors conduct various experiments to evaluate their method.

**Strengths:**

- The quantitative and qualitative results are promising.
- Novel decomposition of edit velocities into cross-prompt and cross-trajectory.
- Novel starting timestep selection for the editing procedure.

**Weaknesses:**

Major:
- Writing clarity - the paper needs refinement in terms of writing clarity. In its current state, it is quite difficult to understand some parts of the method.

For example:

(a) L262 - the orthogonalized cross-prompt should appear rather than cross-trajectory. Then $\Delta_{ts}$ and $\Delta_{ss}$ are defined again, and $V\' \_{edit}(t)$ is defined without any further explanation as well as the hyperparameter $\omega$.

(b) The low frequency and high frequency bands are not defined and are used both in Sections 3.2 and 3.3.2 without any further explanation.

(c) In Eq. (9) $\Delta_{t_i}$ is first presented without any further explanation.

(d) It’s not clear what happens to the channel dimension during the transformation in Eq. (9), and how it translates to the channel dimension of the weighting coefficients defined in Eq. (11).

(e) In Section 3.4 the definitions of $A_j$ and $B_{\phi(j)}$ are not clear - from which attention map calculations are they taken (for example - from the attention maps of $V(X_{tar}, t, c_{src})$?

(f) In Section 4.1.2 - The paragraph is not complete, so it is not fully clear how the text metrics are evaluated.
- Qualitative concerns - some results do not fully follow the target text-prompt, for example in Figure 1 in the ‘cartoon style’ edit - the result is not really a cartoon. In Figure 16 in the ‘fox roses’ edit - the flowers have not really changed into roses.
Moreover, some results do not preserve the structure or identity of the subject, as in Figure 1 in the ‘wearing earrings’ edit - the facial features are changed.

Minor:
- References to the appendix are missing, e.g. in L392.
- $V_{tar}^{src}$, $V_{tar}^{tar}$ and $V_{src}^{src}$ are not defined in the main paper (Figure 3 and 4).
- L296 - Figure number is not mentioned (referenced), and the following paragraph is not fully clear.
- L309 - cross-attention (CA) abbreviation is used before, this abbreviation should be defined earlier.
- L316 - $t$ is not consistently defined. In Eq. (1) it is defined in the range [0, 1], but in Eq. (13) it is defined as an integer indicating the timestep index.
- Figure 7 - Typo in Stableflow.
- Figures 4, 6, 8 - High negative vspace values used.
- Figure 14 - looks very compressed with major artifacts (for example in the sixth and seventh rows).

**Questions:**

- In the Introduction section the authors mentioned that the SPO module determines the editing starting point by calculating the low frequency MSE between the source and target images. However, in Section 3.2 the authors measured the cosine-similarity between the velocity predictions. Which one was used to determine the starting timestep?
- L234-235 - After the “Chaotic Phase” the image information guides the predictions. What happens if from this point we use an empty prompt or another prompt - It would be interesting to see how it affects the generated image path (as in Figure 3).
- L272-274 - It could be beneficial to see a few qualitative examples of the effect of different $\omega$ values.
- The motivation for the editing velocity decomposition is vague. A visualization of these velocities could be beneficial to understand what they represent across different timesteps. Additionally, how do the orthogonal and non-orthogonal components look like?
- L308 - How is the mapping $\phi(j)$ defined? Is it done automatically, or manually chosen by the user? If a token in the source prompt exists in the target prompt, but in another context, what would happen?
For example:
Source prompt - A cat is jumping in a grassy field.
Target prompt - A cat is sitting in a grassy field, and a man is jumping in the background of the scene.
- Section 4.2.2 - It seems like the quantitative evaluation is made with the hyperparameters provided in the official github implementations. However, for different methods, such as RF-Inversion and FireFlow, they provide several hyperparameters. How did the authors choose the set for their evaluations and comparisons?

Please also see weaknesses section.

---

### Official Review · Reviewer_3kLx · 2025-10-31

**Soundness:** 3
**Presentation:** 2
**Contribution:** 2
**Rating:** 4
**Confidence:** 4

**Summary:**

This paper addresses the challenge of high-fidelity, text-guided image editing, particularly in complex, multi-object scenes, using Rectified Flow models. The authors argue that existing methods struggle with a trade-off between structural consistency and semantic alignment  often leading to artifacts or weak edits.  To solve this problem, the authors design a Starting Point Optimization (SPO) strategy and a Trajectory Optimization (TO) strategy. Extensive experiments are conducted to validate their method.

**Strengths:**

1. The paper presents a very well-rounded solution that tackles the editing problem from multiple, complementary angles: SPO, time-domain TO, frequency-domain TO, and Attention Remapping. This multi-pronged approach makes the methodology very solid.

2. The analytical experiments and comparisons with other methods are very thorough, and the results are quite convincing.

**Weaknesses:**

1. The motivation of some strategies is not clearly stated. For example, in Section 3.3 Trajectory Optimization, why are the four terms in Eq. (6) combined into cross-prompt and cross-trajectory?

2. The meaning of some symbols and concepts is not explained. For example,  $V_{src}^{src}$, $V_{tar}^{tar}$ in Figure 3, appearance leakage in Line 73, and so on.

3.  The motivation in the introduction is confusing. The authors base their method on FlowEdit, but don't explain why FlowEdit is unable to perform multi-object editing. Instead, in Lines 65-75, they attribute the difficulty of multi-object editing to the image->latent->image editing framework, which is exactly the problem FlowEdit solves.

4.  The captions of the figures are difficult to understand. For example, it is hard from Figure 5 to conclude the problem in Lines 272-273.

**Questions:**

1. How to obtain $X^{target}_t$ during the editing process? Could the authors provide an algorithm to illustrate the editing process?

2. How does the SPO strategy work? This is not reflected in the main text or Figure 2.

3.  In Section 3.3.2,  why can controlling the frequency components help with the edit? It is hard to understand the motivation.

---

### Official Review · Reviewer_dLgp · 2025-11-01

**Soundness:** 2
**Presentation:** 1
**Contribution:** 3
**Rating:** 2
**Confidence:** 5

**Summary:**

This paper presents a framework for text-guided image editing using rectified flow models, focusing on multi-object and complex scene edits. The method introduces three key components: Starting Point Optimization (SPO), Trajectory Optimization (TO), and attention remapping in MM-DiT. While the approach shows promising results and provides insights into frequency-domain analysis and attention mechanisms, the paper suffers from significant clarity issues that hinder understanding and reproducibility.

**Strengths:**

1. The paper offers a detailed analysis of frequency components during the denoising process (e.g., chaos, layout, and refinement phases) and visualizes attention in MM-DiT layers. This provides valuable insights for future work on rectified flow models, especially in understanding how different stages handle structural and semantic information.
2. The method achieves promising performance on multiple benchmarks.

**Weaknesses:**

1. Poor Writing, which makes it difficult to follow. Many critical elements are not clearly explained, which significantly undermines the clarity of the motivation and main ideas.
   - Section 3.2 (Observations on Phased Editing): This should be the most important section, as its analysis motivates the subsequent method design. However, it is severely lacking in clarity: 1) The three variables V in Figure 3 are not properly defined. How are they calculated? What do the superscripts and subscripts represent? 2) It is unclear what is being visualized, is it the V vectors directly, or the latent representations from a denoising process based on V?
   - SPO Strategy: The details and concrete implementation of the Starting Point Optimization (SPO) strategy are vague. 1) Is a specific step (e.g., step 10, as mentioned in the supplement) used for all images? If so, how does this align with the claim of being "adaptive" ? 2) Where does the adaptivity based on "structural complexity" come from? Does it require a pre-computation (e.g., denoising and similarity calculation) for each individual image? 3) These crucial questions are all missed, addressed only in a single sentence (lines 236-237).

2. The motivation for the Trajectory Optimization (TO) is not well-justified. Why is it necessary to decompose the editing direction into a cross-prompt term and a cross-trajectory term? What is the purpose and theoretical justification for the subsequent orthogonal projection of these terms?

3. The method involves a highly detailed, architecture-specific analysis of the MM-DiT's attention layers, but does not discuss or demonstrate the generalization capability. It remains unclear whether the method, particularly the attention layer analysis and injection strategy, can be transferred to other DiT models, such as Stable Diffusion 3 (SD3).

4. The experimental section lacks a comparison with a key and relevant baseline: ParallelEdits from the PIE-Bench++ (Huang et al., 2025) benchmark. This omission weakens the comprehensiveness of the experimental validation.

**Questions:**

1. How many denoising steps are used?
2. After incorporating additional attention injection, what are the memory usage and inference time when processing a single sample?

---

### Note · Authors · 2025-11-13

I have read and agree with the venue's withdrawal policy on behalf of myself and my co-authors.